# Acetazolamide as an Add-on Therapy Following Barbed Reposition Pharyngoplasty in Obstructive Sleep Apnea: A Randomized Controlled Trial

**DOI:** 10.3390/life14080963

**Published:** 2024-07-31

**Authors:** Simon Hellemans, Eli Van de Perck, Dorine Van Loo, Johan Verbraecken, Scott A. Sands, Ali Azarbarzin, Marijke Dieltjens, Sara Op De Beeck, Anneclaire Vroegop, Olivier M. Vanderveken

**Affiliations:** 1Faculty of Medicine and Health Sciences, University of Antwerp, Wilrijk, 2610 Antwerp, Belgium; 2ENT, Head and Neck Surgery, Antwerp University Hospital, 2650 Edegem, Belgium; 3Multidisciplinary Sleep Disorders Centre, Antwerp University Hospital, 2650 Edegem, Belgium; 4Department of Pulmonology, Antwerp University Hospital, 2650 Edegem, Belgium; 5Division of Sleep and Circadian Disorders, Brigham and Women’s Hospital and Harvard Medical School, Boston, MA 02115, USA; 6Special Dentistry Care, Antwerp University Hospital, 2650 Edegem, Belgium

**Keywords:** acetazolamide, loop gain, OSA, pharmacotherapy, pharyngoplasty, upper airway surgery

## Abstract

Surgical interventions, like barbed reposition pharyngoplasty (BRP), are a valuable alternative for patients with obstructive sleep apnea (OSA) who are unable to tolerate continuous positive airway pressure (CPAP). However, predicting surgical success remains challenging, partly due to the contribution of non-anatomical factors. Therefore, combined medical treatment with acetazolamide, known to stabilize respiratory drive, may lead to superior surgical results. This double-blind, parallel-group randomized controlled trial evaluates the efficacy of acetazolamide as an add-on therapy to BRP in OSA. A total of 26 patients with moderate to severe OSA undergoing BRP were randomized to receive either acetazolamide or placebo post-surgery for 16 weeks. The group who was treated with BRP in combination with acetazolamide showed a reduction in AHI of 69.4%, significantly surpassing the 32.7% reduction of the BRP + placebo group (*p* < 0.01). The sleep apnea-specific hypoxic burden also decreased significantly in the group who was treated with BRP + acetazolamide (*p* < 0.01), but not in the group receiving BRP + placebo (*p* = 0.28). Based on these results, acetazolamide as an add-on therapy following BRP surgery shows promise in improving outcomes for OSA patients, addressing both anatomical and non-anatomical factors.

## 1. Introduction

Obstructive sleep apnea (OSA) is characterized by recurrent episodes of partial (hypopnea) or complete (apnea) upper airway collapse during sleep, leading to intermittent hypoxemia and arousals from sleep. If left untreated, OSA has been associated with adverse cardiovascular, metabolic, neurocognitive and behavioral sequelae [1,2,3].

The pathophysiology of OSA is multifactorial. Upper airway anatomy and collapsibility play a fundamental role. However, in addition to this anatomical predisposition, nonanatomic traits including ventilatory control instability, low respiratory arousal threshold and poor pharyngeal muscle responsiveness also contribute to OSA [4].

Continuous positive airway pressure (CPAP) is generally considered as the first-line treatment for OSA. Although CPAP shows high efficacy, poor adherence remains a critical problem [5,6]. In patients who are unable to tolerate CPAP or other first-line treatments, such as mandibular advancement device (MAD) treatment, surgical interventions may be considered. Notably, upper airway surgery offers an advantage as it eliminates the risk of treatment non-adherence [7]. A large variety of surgical techniques to address obstruction at the level of the soft palate and the lateral pharyngeal walls have been introduced in the last decades [8,9,10,11,12]. Among these, barbed reposition pharyngoplasty (BRP) has gained interest in recent years [13]. In short, this technique uses knotless bidirectional absorbable sutures to expand both the oropharyngeal inlet and the retropalatal space by repositioning the posterior pillar to a more lateral and anterior location [11].

Despite careful patient selection for surgery, including the use of drug-induced sleep endoscopy (DISE), predicting treatment success remains challenging, and surgical outcomes are variable [14,15]. This variability in surgical efficacy is likely attributed to the contribution of the non-anatomical traits in the pathogenesis of OSA. Previous studies have demonstrated that ventilatory control instability (high loop gain) is a predictor of poor surgical response [16,17]. A potential approach to reduce the loop gain and thereby improve surgical outcomes is the administration of acetazolamide [18]. Acetazolamide, a carbonic anhydrase inhibitor, stabilizes respiratory drive by inducing metabolic acidosis [19]. It has been demonstrated that it can reduce loop gain by nearly half in patients with OSA [20].

Taken together, combining acetazolamide with surgical interventions could potentially improve treatment success. Therefore, this study evaluates the efficacy of acetazolamide compared to placebo as an add-on therapy to BRP surgery in patients with OSA.

## 2. Materials and Methods

### 2.1. Study Design

This double-blind, parallel-group randomized controlled trial was designed to assess the efficacy of adjunct acetazolamide compared to placebo in patients with OSA who underwent surgical treatment with barbed reposition pharyngoplasty with or without tonsillectomy (Appendix A). All patients had signed informed consent before surgery. Ethical approval for this study was obtained from the institutional review boards of the Antwerp University Hospital (Belgian registration number: B300201942507). Clinical trial registered with www.clinicaltrials.gov (accessed on 13 June 2024, ID NCT04227093).

### 2.2. Subjects

Patients between the ages of 18 and 75 years with a recent (<2 years) polysomnographic diagnosis of moderate to severe obstructive sleep apnea (apnea–hypopnea index (AHI) between 15 and 65 events/h) were eligible for inclusion. Before the current intervention, positive airway pressure therapy was suggested to all patients. However, they either refused this treatment or stopped using it due to intolerance. Like our routine clinical practice, patients were selected for surgery based on a prior drug-induced sleep endoscopy (DISE). DISE was performed according to the European Position Paper on DISE [21], using a flexible nasopharyngoscope in a semi-dark operating room. A single bolus of midazolam and target-controlled infusion (TCI) of propofol were used for sedation. A primary collapse at the level of the palate or pharyngeal walls was considered a favorable collapse pattern for BRP surgery, while patients with predominantly retrolingual/base of the tongue or epiglottic collapse were not eligible.

Exclusion criteria were as follows: serious psychiatric or neurological disease; body mass index > 35 kg/m^2^; AHI > 65 events/h; central sleep apnea (defined as central AHI ≥ 5 events/h); history of soft palate surgery; craniofacial anomalies affecting the upper airway; unfitness for surgery (American Society of Anesthesiologists (ASA) classification ≥ 3 [22]); contra-indications related to acetazolamide treatment; concomitant intake of drugs that influence breathing, sleep, arousal and/or muscle physiology; pregnancy or willing to become pregnant.

All participants were enrolled at the Department of Otorhinolaryngology, Head and Neck Surgery of the Antwerp University Hospital. At enrolment, the baseline assessment of all patients consisted of a complete medical history and clinical examination, including body mass index (BMI), tonsil score and fiberoptic laryngoscopy.

### 2.3. Randomization

Stratified randomization was performed using web-based software (Qminim, Available at: http://www.scirp.org/journal/PaperInformation.aspx?PaperID=8518, accessed on 13 June 2024) to account for baseline AHI and BMI as potential confounders. Randomization was performed by an independent researcher (D.V.L.) who was not involved in treatment and patient selection. Both patient and clinician were blinded to treatment allocation.

### 2.4. Intervention

BRP with or without tonsillectomy was performed between March 2020 and January 2023. The standardized technique developed by Vicini et al. was used [11]. All BRP procedures were conducted by the same experienced senior surgeon specialized in OSA surgery (A.V.).

Following a three-week recovery period after surgery, patients were randomized to receive either acetazolamide or placebo. Medication was administered as a total daily dosage of 500 mg, split into morning and evening doses of 250 mg each, for 16 weeks. If side effects hindered proper compliance, the dosage could be reduced to a single dose in the evening, subject to approval from the coordinating physician.

### 2.5. Outcome Measurement

The therapeutic response of combination therapy (BRP + medication) was assessed three months after surgery by an in-laboratory polysomnography (PSG). Both baseline and follow-up PSG were performed using standard sleep study equipment and were manually scored according to the American Academy of Sleep Medicine (AASM) guidelines by two experienced technicians blinded to treatment allocation [23,24]. PSG standard protocol recording channels comprise respiratory data (nasal pressure and thermistor), thoracoabdominal movements, pulse oximetry, electroencephalography (EEG), electrooculography (EOG), electromyography (EMG), electrocardiography (ECG), body position and snoring. Apneas were defined as instances of airflow reduction of ≥90% lasting for a minimum of 10 s. Hypopneas were defined as airflow decreases of ≥30% for at least 10 s, accompanied by either a ≥3% oxygen desaturation or EEG arousal. The oxygen desaturation index (ODI) was determined by calculating the number of oxygen desaturations of ≥3% per hour of sleep.

The change in AHI was utilized as the primary outcome. Therapeutic success was defined according to Sher’s criteria; that is, the achievement of a postoperative AHI of less than 20 events per hour of sleep and a ≥50% reduction in preoperative AHI (calculated as ([AHI baseline - AHI follow-up]/AHI baseline) × 100) [25]. Secondary outcomes included the ODI and the percentage of cumulative time with oxygen saturation below 90% in total sleep time (T90). We also quantified sleep apnea-specific hypoxic burden (SASHB), defined as the total area under the respiratory event-related desaturation curve (normalized by sleep time), as previously described [26].

Questionnaires were completed twice during the study: once preoperatively at the time of enrollment and once postoperatively when participants were under medication. These questionnaires include the Epworth Sleepiness Scale (ESS) for determining daytime sleepiness (score range 0 to 24, with higher scores indicating more severe daytime sleepiness), a Functional Outcomes of Sleep Questionnaire-10 (FOSQ-10; score range 5–20 with higher scores indicating better functional outcome), and a visual analog scale (VAS) for snoring (score range 0–10, with higher scores indicating increased snoring loudness). Side effects were assessed during the study and quantified with a VAS.

Additionally, OSA endotypes were calculated and analyzed as part of an exploratory analysis. These calculations were performed using previously validated methods on baseline and follow-up clinical polysomnography data [27,28,29]. The calculated traits include loop gain, respiratory arousal threshold, upper airway collapsibility and pharyngeal muscle responsiveness/compensation.

### 2.6. Statistical Analysis

Data collection was conducted using the OpenClinica open-source software, version 3.3 (OpenClinica, LLC, Waltham, MA, USA). Statistical analysis and data management were performed using software packages JMP Pro software (version 16.0, SAS Institute Inc., Cary, NC, USA). Endotypes and SASHB were calculated using the previously validated algorithm in MATLAB (MATLAB and Statistics Toolbox Release 2022a; Mathworks, Natick, MA, USA) [26,27,28,29]. Descriptive statistics were presented as medians and 1st and 3rd quartiles [Q1–Q3]. A Mann–Whitney U test was used to compare groups. The Wilcoxon test was used for intragroup comparisons. Exploratory analyses were conducted using univariate and multivariate regression analysis. To assess the impact of outliers on the data, an outlier analysis was performed. Analyses were performed both with and without outliers. Probability values lower than 0.05 were considered statistically significant.

## 3. Results

A total of 26 patients were randomized to receive either BRP + acetazolamide or BRP + placebo. The baseline characteristics of both groups after randomization are summarized in Appendix A. Statistical analysis revealed no significant differences between the two groups at baseline. Of the 26 patients who participated in the study, 21 completed the trial (9 with acetazolamide and 12 with placebo, as shown in Figure 1). There were no intra-operative complications. One patient was readmitted for pain management, administration of fluids and nutrition. Otherwise, there were no complications related to surgery. Three patients (all in the acetazolamide group) reported disturbing side effects (paresthesia, taste disturbances and muscle cramps), requiring dose adjustment to 250 mg once a day in the evening. No significant difference in AHI reduction was found for patients with the reduced dose compared to patients receiving the full dose. No severe adverse events (SAE) occurred during the study.

Patients were generally overweight, middle-aged males with a median baseline AHI of 23.6 (19.9–35.7) events/h (Table 1). The median time between the baseline PSG and surgery was 211(82–318) days. In both groups, medication treatment was discontinued by one patient due to side effects. In the acetazolamide group, three patients chose not to participate in the study anymore after recovering from the surgery and therefore did not initiate the study medication (Figure 1).

Outlier analysis (using a tail quantile of 0.1 and a Q of 3) conducted on percent reduction in AHI identified a single outlier within the placebo group (age: 45 years, BMI: 30.7 kg/m^2^, delta AHI: 388%). In a further analysis of this patient’s data, the cause for this substantial increase could not be identified. Moreover, the patient’s fatigue symptoms remained unchanged after surgery. Analyses were performed with and without the removal of this outlier. The main paper describes results with the outlier included. The outlier was excluded only for exploratory analysis of endotypes. Analyses of the main outcomes without the outlier can be found in the Appendix A.

### 3.1. Primary Outcome

In patients who were treated with BRP in conjunction with supplementary acetazolamide treatment, there was a reduction in the AHI from 25.2 [20.6–47.8] at baseline to 7.7 [5.3–14.5] events/hour at follow-up (*p* < 0.01). In the group that was treated with BRP along with placebo, the AHI reduced from 21.7 [19.5–34.8] at baseline to 15.0 [10.8–32.1] events/hour (*p* = 0.11) at follow-up (Figure 2). The median reduction in preoperative AHI was significantly higher in the acetazolamide group when compared to placebo (69.4% [59.9–77.7] vs. 32.7% [11.5–49.7] reduction from baseline respectively, *p* < 0.01). Out of 9 patients in the acetazolamide group, 8 (89%) showed a treatment response according to Sher’s criteria [25], compared to 3 out of 12 patients (25%) in the placebo group.

### 3.2. Secondary Outcomes

In the BRP + acetazolamide group, both the T90% (*p* = 0.02) and ODI (*p* < 0.01) significantly decreased (Table 2). However, no significant changes in these parameters were observed in the BRP + placebo group. The hypoxic burden decreased significantly in the group who was treated with BRP + acetazolamide (36.3 [26.4–62.2] to 10.3 [7.9–18.0] %min/h, *p* < 0.01), but not in the group receiving BRP + placebo (32.9 [21.9–62.9] to 31.2 [15.7–51.9] %min/h, *p* = 0.28).

Although the ESS decreased in both treatment groups post-intervention, the reduction did not reach statistical significance (Table 2). Notably, approximately half of the patients (*n* = 10) were sleepy at baseline (i.e., ESS > 10), comprising three patients in the acetazolamide group and seven in the placebo group. When considering only the baseline sleepy patients (ESS > 10), the ESS decreased significantly from 15 (11.8–16.0) to 10 (3.0–14.5), *p* = 0.02, with four patients (two in each treatment group) still reporting excessive daytime sleepiness at follow-up.

The VAS for snoring showed a significant reduction in the group treated with BRP with acetazolamide, while no significant change was observed in the group receiving a placebo as an adjunct therapy (Table 2). At follow-up, 10 patients (seven (78%) in the BRP + acetazolamide group and 4 (33%) in the BRP + placebo group) reported that snoring was only mild and no longer socially disturbing (VAS ≤ 3).

### 3.3. OSA Endotypes

The OSA endotypes were also calculated as part of an experimental analysis. For these analyses, the outlier was excluded. Except for a decreasing arousal threshold in the acetazolamide group, no significant difference was found between baseline and follow-up for all endotypes (Table 3, Appendix A). In the placebo group, the baseline loop gain was associated with the reduction in AHI (r^2^: 0.52, *p* = 0.02). Even after accounting for age, BMI and baseline AHI, this association remained statistically significant (*p* = 0.03). However, this association was not observed in the acetazolamide group (Figure 3).

## 4. Discussion

The present randomized controlled trial was designed to determine the effect of acetazolamide as an add-on therapy to upper airway surgery in patients suffering from moderate to severe obstructive sleep apnea. The findings of our study demonstrated that patients receiving combination therapy achieved significantly better outcomes than those undergoing surgery alone.

In the group receiving combination therapy with acetazolamide, all but one individual met the responder criteria based on AHI. Considering that a significant portion of patients continue to experience residual OSA following surgery, this finding is highly valuable [30,31].

In addition to the standard polysomnographic metrics, SASHB was also calculated. This is a newer metric that appears to be a better measure of OSA severity concerning the cardiovascular disease consequences [26,32]. Within the present study group, the decrease in SASHB was significantly greater in the group treated with acetazolamide as an adjunct. This finding suggests that combining BRP with acetazolamide might enhance cardiovascular protection compared to surgery alone.

Two small studies have previously explored the efficacy of acetazolamide as an adjunctive therapy for upper airway surgery. Inoue et al. observed in five patients that combination therapy of uvulopalatopharyngoplasty (UPPP) and acetazolamide resulted in more pronounced reductions in AHI compared to UPPP and acetazolamide used as monotherapies [33]. Likewise, Vanderveken et al. observed positive results in patients with mild OSA when comparing UPPP combined with acetazolamide to UPPP alone [34]. Similar to acetazolamide, recent research has demonstrated the beneficial results of oxygen in patients who failed to respond to surgery [35]. These findings are consistent with ours, indicating that treatments targeting loop gain reduction could be valuable additions to upper airway surgery.

Previous studies on acetazolamide for OSA have shown promising results. However, a potential issue lies in its tolerability, which could hinder its long-term use [36]. Despite this concern, long-term outcomes are not yet available [37]. To the best of our knowledge, the current study has, with 16 weeks, the longest follow-up of acetazolamide for obstructive sleep apnea so far. Out of the patients who received acetazolamide in this trial, 33% required a dose reduction due to side effects, and one patient discontinued the therapy. Similarly, in the placebo group, medication was discontinued by one individual due to adverse effects. Despite the need for dose reduction in some patients, overall tolerance was reasonably good, with side effects typically limited to paresthesias and dysgeusia. No serious adverse events occurred during the current study. Furthermore, the reduction in AHI was comparable between patients who received 500 mg and those who reduced their dose to 250 mg (%reduction in AHI: 66.2% (56.0–90.4) with 250 mg vs. 70.8% (55.6–76.8) with 500 mg; *p* = 0.99). So, it might be feasible to initiate acetazolamide at a lower dose and gradually increase the dose if needed. However, more research is needed.

Non-anatomical pathophysiologic traits contribute to OSA. Some of these traits, especially high loop gain, are associated with suboptimal surgical outcomes and thereby contribute to the unpredictability of surgical interventions for OSA [17]. This is in line with our finding that a high loop gain was inversely associated with the reduction in AHI in the placebo group. Patients with a high baseline loop gain experienced less pronounced reductions in AHI following surgery. However, this association was not observed in patients who received acetazolamide as an add-on therapy, potentially suggesting that combination therapy of surgery with acetazolamide is beneficial for patients with a high baseline loop gain. Therefore, our study’s findings support the notion that addressing both anatomical and non-anatomical traits leads to superior outcomes.

Contrary to our expectations, there was no significant reduction in loop gain observed in the acetazolamide group, despite an earlier study by Edwards et al. proposing a 41% decrease with acetazolamide [20]. This disparity in findings can be attributed to various factors. Firstly, the dosage of acetazolamide in the Edwards et al. study was twice as high as the dosage used in our research. A higher dosage may be needed to achieve more substantial reductions in loop gain. Secondly, our study administered acetazolamide for 16 weeks, whereas previous research on acetazolamide for OSA typically employed short-term interventions [37]. Consequently, the long-term effects of loop gain remain unclear. Nonetheless, our study showcased positive effects on AHI reduction with extended administration of acetazolamide, indicating a favorable outcome. It is plausible that the positive effects of acetazolamide extend beyond merely reducing loop gain, and that other mechanisms such as fluid shift contribute to an improved outcome [38].

An important limitation of this study was the lack of a postoperative PSG before starting medication. Given the variability in surgical outcomes, conducting an additional interim PSG would have allowed confirmation that both groups remained comparable before starting medical therapy and would have provided a clearer assessment of improvement with medication. However, because treatment response can only be reliably measured several months after surgery, conducting an additional interim PSG would significantly prolong the follow-up period.

This study is also limited by the fact that it did not use a crossover design, where each patient would receive both a placebo and acetazolamide after surgery. Instead, we opted for a parallel design with an extended follow-up period. This approach allowed for prolonged acetazolamide administration and facilitated the assessment of its feasibility during extended intake, an aspect that has not been thoroughly investigated in previous studies.

Another limitation is the limited sample size, which can potentially affect the generalizability of the observed results. Moreover, the asymmetric dropout rate observed in our study, with four patients dropping out from the acetazolamide group and one from the placebo group, could introduce a potential bias. This imbalance might skew the findings, as the dropouts may reflect differences in treatment tolerability or efficacy. Despite this constraint, it is essential to emphasize that our study serves as a pilot investigation, laying the groundwork for subsequent research in this field.

In summary, this study illustrates that combining acetazolamide with upper airway surgery significantly enhances outcomes for patients with moderate to severe obstructive sleep apnea. Combination therapy resulted in a more substantial reduction in AHI and SASHB compared to surgery with a placebo adjunct. However, acetazolamide did not show any impact on loop gain in our study population. Given the promising results of combination therapy in managing residual OSA after surgery, further investigation through larger, more comprehensive studies is warranted.

## Figures and Tables

**Figure 1 life-14-00963-f001:**
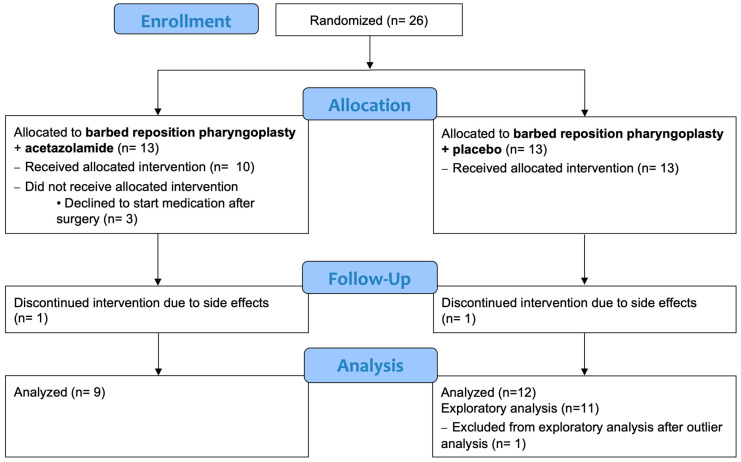
Consort diagram.

**Figure 2 life-14-00963-f002:**
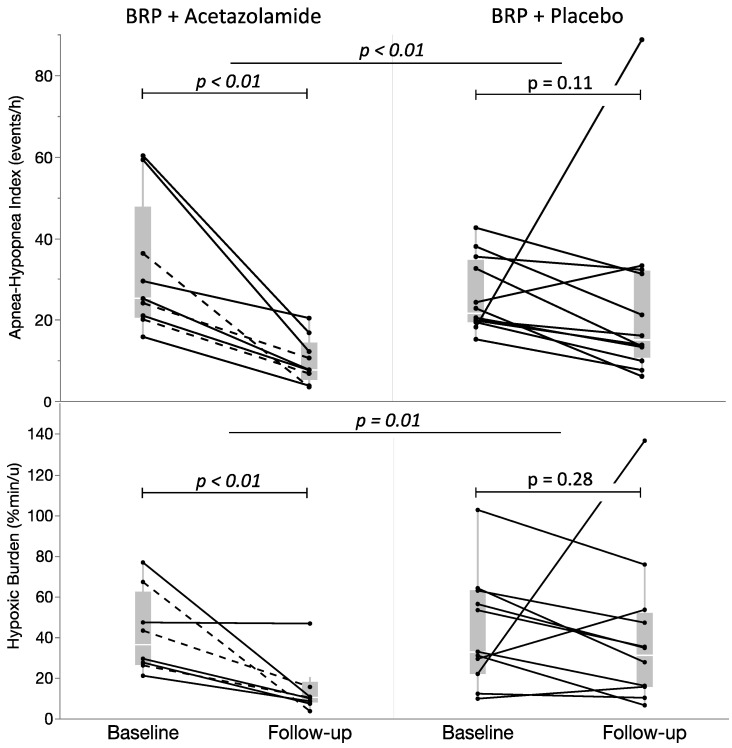
Apnea-hypopnea index (**above**) and sleep apnea-specific hypoxic burden (**below**) before and after treatment (BRP + acetazolamide (**left**) and BRP + placebo (**right**)). Dotted lines represent patients for whom medication dose was reduced to a single dose each day. Calculation of hypoxic burden was not possible for two patients (one in each treatment arm) due to the unavailability of raw PSG data. Abbreviations: BRP: barbed reposition pharyngoplasty.

**Figure 3 life-14-00963-f003:**
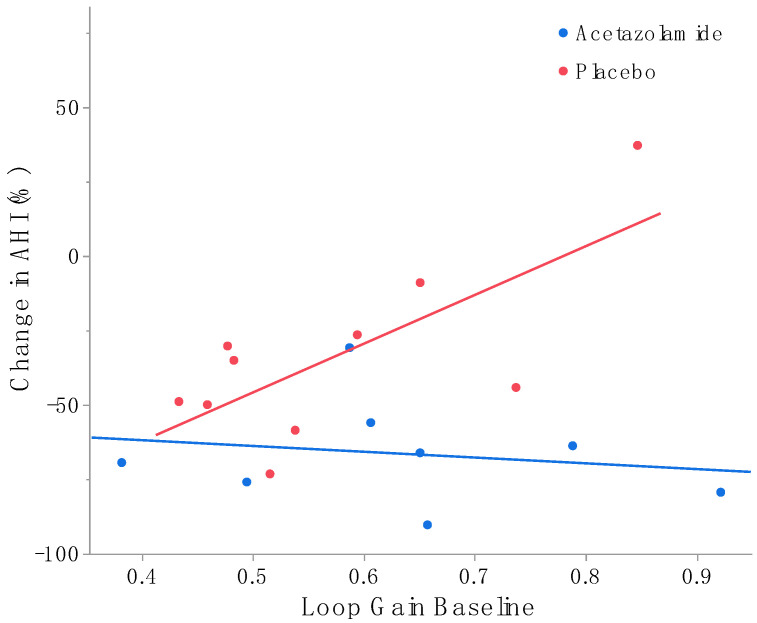
Linear regression showing association between baseline loop gain and the change in AHI after treatment. A negative delta AHI indicates a decrease in AHI after treatment. Baseline loop gain is associated with the reduction in AHI in the placebo group, indicating that at a high baseline loop gain, there are less pronounced reductions in AHI. However, this is not the case in the group receiving acetazolamide as add-on therapy.

**Table 1 life-14-00963-t001:** Characteristics of all patients.

Study Population (*n* = 26)	
Demographics and anatomical features	
Age, years	46.0 (36.8–60.3)
Gender, male (%)	23 (88.5)
BMI, kg/m^2^	28.5 (26.8–30.0)
Tonsil grade, No. (%)	
0	3 (11.5)
1	14 (53.8)
2	6 (23.1)
3	2 (7.7)
4	1 (3.8)
Mallampati grade, No. (%)	
1	4 (15.4)
2	9 (34.6)
3	5 (19.2)
4	8 (30.8)
Baseline polysomnographic parameters	
TST, min	425.5 (370.5–453.0)
SEI, %	86.1 (74.7–91.6)
Mean SpO_2_, %	94 (93.1–95.1)
T90%, %TST	2.3 (0.4–7.9)
ODI_3_, events/h	21.5 (15.2–29.7)
AHI, events/h	23.6 (19.9–35.7)
SASHB, (%min/h)	43.2 (26.4–63.8)

Values presented as median (Q1–Q3) or number (%). The patients who dropped out of the study exhibited similar clinical and baseline polysomnographic characteristics as the other patients. Abbreviations: BMI = body mass index; TST = total sleep time; SEI = sleep efficiency index; SpO_2_ = oxygen saturation; T90 = sleep time with oxygen saturation ≤ 90%; ODI = oxygen desaturation index; AHI = apnea–hypopnea index; SASHB = sleep apnea-specific hypoxic burden.

**Table 2 life-14-00963-t002:** Baseline and follow-up polysomnographic results and questionnaire scores of both treatment groups.

	BRP + Acetazolamide (*n* = 9)			BRP + Placebo (*n* = 12)		
	Baseline	Follow-Up	*p*	Baseline	Follow-Up	*p*
Sleep characteristics						
TST, min	424 (356–446)	414 (408–454)	0.426	410 (368–453)	424 (416–450)	0.129
SEI, %	82.8 (73.5–89.6)	84.7 (81.7–90.5)	0.426	84.0 (73.2–94.0)	87.1 (84.9–89.5)	0.147
REM, %TST	20.2 (14.7–23.3)	24.5 (18.0–30.2)	0.059	20.1 (17.0–21.6)	20.4 (16.3–26.6)	0.176
N1, %TST	11.5 (5.0–16.0)	5.2 (3.0–9.5)	0.020	12.4 (2.1–19.6)	8.7 (4.0–12.6)	0.240
N2, %TST	61.8 (46.6–65.1)	51.6 (49.0–60.9)	0.129	54.3 (46.0–61.3)	55.9 (50.6–60.0)	0.309
N3, %TST	11.7 (6.8–21.3)	17.2 (12.1–26.0)	0.164	15.7 (3.2–21.0)	15.8 (9.1–21.1)	0.229
Mean SpO_2_, % (*)	94.1 (92.4–95.5)	95.2 (93.8–95.7)	0.008	94.0 (93.4–95.1)	93.9 (92.9–94.4)	0.373
Nadir SpO_2_, %	86 (83.5–89.0)	89.0 (87.3–90.8)	0.219	84.5 (81.5–88.3)	86.1 (82.5–88.0)	0.531
T90%, %TST (*)	2.3 (0.3–7.8)	0.2 (0.0–3.3)	0.016	1.8 (0.2–3.7)	1.6 (0.7–6.5)	0.451
ODI_3_, events/h (*)	22.1 (12.2–39.0)	9.0 (5.6–15.0)	0.004	17.9 (13.3–25.7)	16.6 (9.6–26.1)	0.622
AHI, events/h (*)	25.2 (20.6–47.8)	7.7 (5.3–14.5)	0.004	21.7 (19.5–34.8)	15.0 (10.8–32.1)	0.110
AHIsupine, events/h	51.6 (36.9–67.3)	27.5 (9.1–53.2)	0.074	47.5 (38.8–69.2)	33.6 (17.1–56.9)	0.266
AHInonsupine, events/h	19.1 (2.3–29.0)	4.1 (2.0–8.8)	0.055	9.1 (1.1–17.8)	7.5 (3.5–26.6)	0.684
AI, events/h	1.8 (0.5–6.8)	0.2 (0.1–0.8)	0.008	1.3 (0.0–3.4)	0.5 (0.0–4.6)	0.920
HI, events/h (*)	23.3 (20.0–33.2)	7.4 (5.0–13.8)	0.004	18.8 (17.5–32.4)	14.7 (8.4–29.0)	0.151
OAHI, events/h (*)	23.9 (20.5–46.6)	7.6 (5.0–13.8)	0.004	20.9 (18.4–34.6)	14.7 (8.1–31.2)	0.110
CAHI, events/h	0.2 (0.0–1.3)	0.3 (0.0–0.8)	0.219	0.2 (0.0–0.9)	0.2 (0.0–1.2)	0.773
SASHB, (%min/h) (*)	36.3 (26.4–62.2)	10.3 (7.9–18.0)	0.008	32.9 (21.9–62.9)	31.2 (15.7–51.9)	0.278
Patient-reported symptom scores						
ESS	8.0 (4.0–15.5)	5.0 (2.0–10.5)	0.164	11.0 (5.5–14.8)	6.0 (3.0–10.0)	0.113
FOSQ-10	18.0 (14.1–19.5)	19.4 (18.1–20.0)	0.102	15.4 (11.8–18.1)	17.7 (15.0–18.8)	0.051
VAS snoring	8.0 (6.3–9.8)	3.0 (1.5–4.0)	0.008	8.0 (6.0–10.0)	4.0 (3.0–9.0)	0.059

The *p*-values for before–after comparisons were determined using the Wilcoxon signed-rank test. A Mann–Whitney U test was conducted to compare the pre–post-treatment differences between the placebo and acetazolamide group for each outcome. The presence of a statistically significant difference between the two groups, indicating that the change after treatment in the acetazolamide group significantly differs from that of the placebo group, is denoted by an asterisk (*****). Abbreviations: TST = total sleep time; SEI = sleep efficiency index; REM = rapid eye movement; N = non-REM sleep stages 1–3; SpO_2_ = oxygen saturation; T90 = sleep time with oxygen saturation ≤ 90%; ODI = oxygen desaturation index; AHI = apnea–hypopnea index; AI = apnea index; HI = hypopnea index; OAHI = obstructive apnea–hypopnea index; CAHI = central apnea–hypopnea index; SASHB = sleep apnea-specific hypoxic burden; ESS = Epworth sleepiness scale; FOSQ-10 = functional outcome of sleep questionnaire; VAS = visual analogue scale.

**Table 3 life-14-00963-t003:** Baseline and follow-up endotypes of both treatment groups.

	BRP + Acetazolamide (*n* = 9)			BRP + Placebo (*n* = 11)		
	Baseline	Follow-Up	*p*	Baseline	Follow-Up	*p*
Vpassive (*)	70.3 (64.3–73.8)	79.8 (76.1–87.0)	0.195	73.8 (67.6–78.3)	74.9 (72.5–78.8)	0.322
Muscle compensation (Vcomp)	0.1 (−8.8–6.3)	3.8 (1.5–4.3)	0.313	1.3 (−3.9–4.7)	3.4 (0.4–3.7)	0.999
Arousal threshold (*)	137.8 (107.2–156.0)	117.8 (100.0–131.1)	0.031	134.9 (109.8–154.0)	119.6 (112.7–131.5)	0.426
LG1	0.62 (0.52–0.76)	0.56 (0.37–0.78)	0.641	0.53 (0.47–0.67)	0.60 (0.48–0.73)	0.375

The *p*-values for before–after comparisons were determined using the Wilcoxon signed-rank test. A Mann–Whitney U test to compare the change in both treatment groups showed no statistically significant differences. * = transformed values. Abbreviations: LG = loop gain.

## Data Availability

The data presented in this study are available upon request from the corresponding author.

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
