# Peer review of "Acetazolamide as an Add-on Therapy Following Barbed Reposition Pharyngoplasty in Obstructive Sleep Apnea: A Randomized Controlled Trial"

_life, 2024, doi:10.3390/life14080963_

Round 1
Reviewer 1 Report
Comments and Suggestions for Authors
The authors of this article present the results of a randomised trial that evaluated the effect of acetazolamide as an add-on therapy to surgical treatment of OSAS. The paper is very interesting and draws the reader's attention to several aspects related to the treatment of sleep apnoea. However, there are some questions and concerns that need clarification.
Subjects
My very significant concern is the fact that there is a very long interval from baseline polysomnography (median 211 days) to surgery. Were other treatments for OSAS undertaken in patients during this time - for example by recommending weight reduction? If patients had a significant change in body weight between the initial PSG and surgery, this fact may have had a significant impact on the severity of OSAS and subsequent assessment of the effect of surgical treatment.
A citation with information on eligibility for surgical treatment according to the American Society of Anesthesiologists should be added to the refrences list.
The section describing patients lacks information on comorbidities and smoking. It is important to add information on the coexistence of heart failure and the relevant treatment. Acetazolamide is a diuretic. Therefore, an additional beneficial effect, which may be a diuretic effect in patients with coexisting heart failure and the effect of reducing venous stasis on the severity of sleep-disordered breathing, cannot be excluded.
It also seems reasonable to provide eligibility criteria for simultaneous tonsillectomy and to present the distribution of patients who underwent tonsillectomy to the subgroup using acetazolamide.
In the description of the polysomnographic recording, there is information about the recording of snoring sounds, but in the results section the authors present only a subjective assessment using VAS. In my opinion, the results of the analysis of snoring sounds from the PSG recording should be presented.
At least a brief description of how loop gain, respiratory arousal threshold, upper airway collapsibility and pharyngeal muscle responsiveness/compensation are calculated should be included in the text.
The description of the statsitical method lacks an explanation of why the authors used the median. The text should include information about the distribution of the data. If the distribution is not normal, the use of median and quartile range is justified.
Discussion
The initial part of this chapter should contain information that in the BR+placebo group, no significant differences were observed in any of the studied parameters, which actually makes one wonder about the effectiveness of this method of surgical treatment. A treatment response according to Sher's criteria in 3 out of 12 patients (25%) is not a very optimistic observation.
Author Response
Thank you for your valuable contribution to improving the manuscript. The notes to revier are included in the attached document.

Reviewer 2 Report
Comments and Suggestions for Authors
Dear authors,
your study analyzed the impact of a combined treatment surgery + acetazolamide versus surgery + placebo, The study design was a randomized trial. Randomization was after surgery using a software. Patients and researchers were blinded.
After randomization, neither group was compared for BMI, PSG data, ESS, snoring, and FOSC, to ensure that both groups were similar in their composition for these outcomes. I suggest including this analysis.
Comparison was made for both groups between baseline and after treatment (improvement within the group), but no comparision between both groups were made.
26 patients were enrolled, 5 dropped out (4 in medication group, 1 in placebo group). As there are only few patients enrolled, this assymetric drop out might be a bias ( mostly looking that 3 declined to use medication probably due to very satisfactory results after surgery), what should be discussed better. The small number of enrolled patients also will influence the effect of the result which should be discussed better and should be considered at conclusion.
PSG was performed before surgery and after 4 months of medication, no intermediate (posop) PSG was performed. As surgical results are variable, it is not possible to confirm that both groups were still similar (considering that there were similar with the preop PSG). This is a major bias and should be pointed out as a weakness of the study.
Acetazolamide showed a higher effect on hypoxic burden, stabilizing oxygen saturation. It did not improve Loop Gain, as hypothesized and discussed by the authors. I think the no-effect on LG should also be mentioned in conclusion. Did you compare oAI and HI ? was there an improvement in Apnea index? Or did you find a better stabilization of hypopneas?
Author Response

(The authors gave the same response as above.)

Round 2
Reviewer 2 Report
Comments and Suggestions for Authors
Dear authors,
the new version enhances the quality of your study, while considering its limitations as a "pilot study". Thank you for improving the text. It is more clear and certainly suitable for publication.